# Adaptive Curriculum Learning for RLHF with Influence-Based Cluster Bandits

## Abstract

Reinforcement learning (RL) plays a central role in post-training large language models (LLMs). Yet, existing RLHF pipelines typically rely on fixed or uniform sampling strategies, which fail to adapt to the model's evolving learning state. This mismatch leads to wasted computation on less informative samples while neglecting instances with higher training impact, ultimately limiting efficiency, generalization, and performance gains. We introduce an adaptive curriculum learning framework that integrates influence-based clustering with a multi-armed bandit (MAB) scheduler. Training data are partitioned into clusters defined by semantic and difficulty-related features, each treated as an arm in the MAB formulation. A Cluster Score (CS), updated via sliding-window influence functions, quantifies the dynamic importance of each cluster as the model evolves. This adaptive scoring drives the scheduler to balance exploitation of high-impact clusters with exploration of underrepresented regions, ensuring efficient learning while maintaining diversity. Unlike prior approaches that overfit to narrow high-reward subsets, our cluster-level sampling prevents redundancy and broadens representational coverage. Experiments with Group Relative Policy Optimization across mathematical reasoning benchmarks show that our method consistently accelerates convergence and improves generalization. These results highlight the value of distribution-level adaptive curricula in advancing RLHF for LLM training.

## 1 Introduction

Reinforcement learning (RL)-based post-training has emerged as a powerful paradigm for large language models (LLMs), particularly in tasks requiring structured reasoning and multi-step inference Ouyang et al. (2022). By leveraging reward signals derived from task performance, human feedback, or domain-specific metrics, RL enables models to optimize directly toward behavioral objectives rather than merely imitating reference outputs. This approach has proven especially effective for complex reasoning and agentic tasks, as demonstrated by recent breakthroughs in reasoning models such as OpenAI-O1 Jaech et al. (2024) and DeepSeek-R1 Guo et al. (2025).

Despite these advances, a fundamental challenge remains underexplored: **how to dynamically schedule training samples across diverse data distributions to maximize learning efficiency**. LLMs are typically post-trained on datasets spanning multiple domains—ranging from factual QA to mathematical problems and coding tasks—each differing in knowledge relevance, difficulty, and alignment objectives. As models evolve during training, their capacity to learn from different distributions changes: what is challenging at one stage may become trivial or overwhelming at another. Without adaptive scheduling, training resources may be wasted on distributions offering diminishing returns, while potentially valuable data remains underutilized.

The most commonly used fixed or uniform sampling strategies fail to account for this evolution, leading to two key limitations. (1) Lack of adaptive sample scheduling: Most approaches either sample uniformly or rely on fixed curricula, ignoring the model's evolving learning state. This prevents the training process from prioritizing distributions that are most informative at each stage, reducing training efficiency and slowing performance improvement. (2) Limited diversity in high-value sample selection: Methods that prioritize samples solely based on quality metrics, such as difficulties, tend to select similar instances. This concentration can bias the model toward specific patterns, neglecting rare but informative cases and hindering generalization. These limitations, while

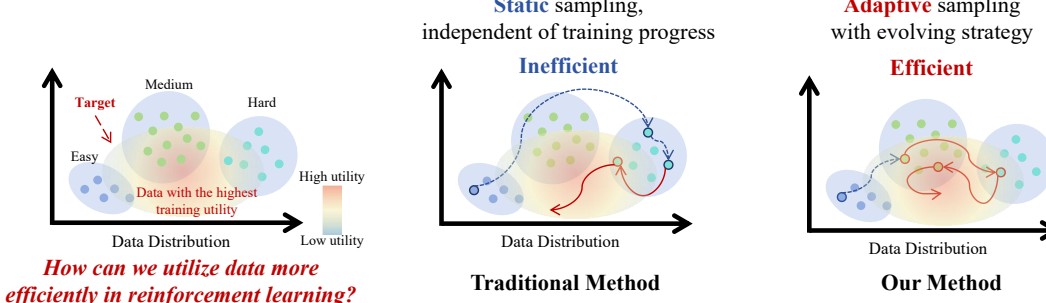

Figure 1: A region of high-utility data exists in the data distribution that makes model training more efficient. Traditional random sampling methods fail to identify and focus on this region. Our method leverages influence function to dynamically measure the changing value of data to the model, and combines it with a multi-armed bandit to balance sampling diversity. This allows us to adaptively identify and utilize efficient training data, improving learning efficiency.

less critical in pretraining or supervised fine-tuning where objectives and feedback are dense and stable, become detrimental in the RLHF stage, where rewards are sparse, noisy, and dynamically shaped by the evolving policy, making existing adaptive scheduling strategies insufficient.

To address these challenges, we propose an influence-function-based multi-armed bandit (MAB) framework for adaptive curriculum learning in RLHF. We first formulate distribution-level curriculum learning as an MAB problem, where each cluster, constructed from a combination of semantic and difficulty-related features, represents an arm. For each cluster, we compute a Cluster Score (CS) by applying influence functions within a sliding window of training steps, using gradient-based estimates to capture its evolving contribution to model updates. The sliding window ensures that scores reflect the model's current learning state rather than being dominated by early dynamics. Based on these scores, the MAB scheduler adaptively balances exploitation of high-value clusters with exploration of underrepresented ones, promoting both efficiency and broad coverage. To further improve scalability, we adopt an efficient influence function computation based on Conjugate Gradient (CG) approximation, allowing practical assessment of cluster-level importance at scale without incurring prohibitive computational cost. In addition, our clustering-based sampling strategy preserves diversity by discouraging over-selection of homogeneous high-scoring instances, while still prioritizing impactful training data. Our work makes the following key contributions:

- We propose a **cluster-level influence function** method to systematically and dynamically estimate the contribution of training samples to parameter updates in RLHF. By computing influence scores within **sliding windows of training steps**, our approach ensures that curriculum design reflects the model's current learning state rather than being dominated by early training dynamics.
- We design a **multi-armed bandit (MAB) scheduler** that adaptively selects clusters throughout training. By integrating influence-based cluster scores with exploration–exploitation trade-offs, our framework dynamically aligns data selection with the model's evolving capabilities.
- We empirically validate our framework on GRPO tasks across multiple mathematical reasoning benchmarks, demonstrating consistent improvements in final performance, faster convergence, and better generalization compared to existing curricula and baselines.

## 2 RELATED WORK

### 2.1 REINFORCEMENT LEARNING FROM HUMAN FEEDBACK

Aligning large language models with human intent is primarily addressed by Reinforcement Learning from Human Feedback (RLHF), a multi-stage paradigm of reward modeling and policy optimization Ouyang et al. (2022); Bai et al. (2022). The inherent complexity of this pipeline has led

to two main research thrusts. The first seeks to simplify the process: Direct Preference Optimization (DPO) recasts alignment as a direct policy optimization problem, bypassing the need for an explicit reward model Rafailov et al. (2023), and VAR further distills this into a reward-weighted supervised loss Du et al. (2025). The second works to refine the existing framework; for example, R3HF improves the granularity of the feedback with token-level rewards Li et al. (2024), while UP-RLHF introduces uncertainty penalties to curb overoptimization Zhai et al. (2023). In parallel, to improve specialized capabilities, such as mathematical reasoning, Group Relative Policy Optimization (GRPO) emerged as a powerful variant of PPO Shao et al. (2024), inspiring a lineage of follow-up methods, including DAPO, GSPO, GMPO, and others, that aim to improve its stability and efficiency Yu et al. (2025); Zheng et al. (2025); Zhao et al. (2025); Chen et al. (2025); Liu et al. (2025); Shrivastava et al. (2025).

A key limitation of these RLHF methods, including GRPO, is the dependence on a uniform or static sampling strategy. This non-adaptive approach leads to inefficient training, as it fails to prioritize data according to the model's evolving capabilities. To address this, we introduce an adaptive curriculum learning framework that uses influence functions and a bandit scheduler to dynamically select the most informative data for the current model state.

## 2.2 MULTI-ARMED BANDIT FOR SAMPLE SELECTION

The multi-armed bandit Vermorel & Mohri (2005) is a powerful framework for sequential decision-making, designed to optimally balance the exploration-exploitation trade-off. It has been used to optimize the LLM inference process, such as dynamic prompt engineering or adaptive response generation. Another paradigm integrates the LLM into the bandit algorithm itself, either by directly using the LLM as a decision-making agent, a method shown to be suboptimal due to poor exploration Krishnamurthy et al. (2024) or, more effectively, by using the LLM as a sophisticated reward predictor within a classical MAB framework Sun et al. (2025). These applications primarily focus on optimizing interactions at inference time or improving the bandit mechanism itself. However, the application of MAB algorithms to dynamic curriculum learning in RLHF remains unexplored. Our work bridges this gap by formulating data cluster selection as a multi-armed bandit problem, where influence functions provide rewards for decision-making.

## 2.3 CURRICULUM LEARNING IN DEEP LEARNING

Curriculum Learning (CL), the strategy of presenting training data in a meaningful "easy-to-hard" order, is a pivotal training paradigm shown to improve model generalization and efficiency Bengio et al. (2009); Zaremba & Sutskever (2014). Consequently, a central research thrust has been the automation of curriculum design, moving beyond manual or heuristic-based orderings Toborek et al. (2025). This has been realized through diverse mechanisms, including the Teacher-Student framework where a "teacher" model selects tasks based on student progress Matiisen et al. (2019); Portelas et al. (2020), unsupervised self-play Sukhbaatar et al. (2017), and the open-ended co-evolution of environments and solutions Wang et al. (2019). However, prior methods often depend on heuristic proxies for sample importance and fail to adapt to the model's evolving state. We instead introduce a cluster-level influence function that dynamically estimates sample contributions within sliding training windows, ensuring curriculum design reflects current learning rather than predefined heuristic rules, while naturally complementing bandit-based exploration–exploitation strategies.

## 3 METHODS

### 3.1 OVERALL FRAMEWORK

We propose an automated distribution-level curriculum learning framework that dynamically adjusts the sampling distribution across different data sources during the RLHF training. Unlike traditional approaches that treat all training data uniformly, our method prioritizes samples based on their estimated impact on model performance, leveraging influence functions and multi-armed bandit algorithms. Figure 2 illustrates the overall architecture consisting of three main components. **Data Clustering Module:** Organizes training samples into semantically and difficulty-coherent clusters, which serve as the distributional units for curriculum learning. **Influence Function Computation:**

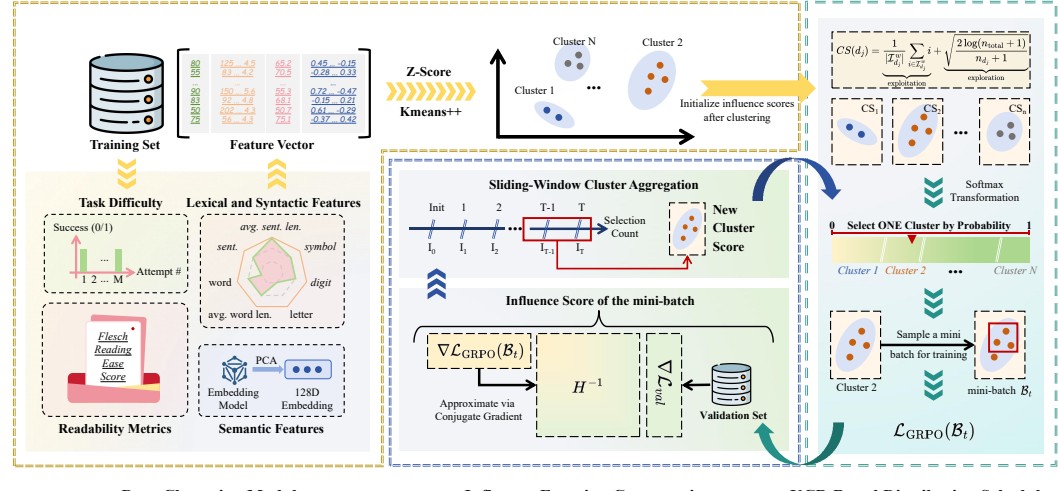

Figure 2: Overall framework of our influence-based adaptive curriculum learning method. Distribution-level influence scores guide a UCB-based scheduler to adaptively adjust sampling probabilities during training.

Measures the learning value of each cluster to guide adaptive sampling. **UCB-Based Distribution Scheduler:** Uses cluster-level scores to balance exploration of underrepresented clusters and exploitation of high-impact clusters. We next describe each component in detail.

## 3.2 DATA REPRESENTATION AND CLUSTERING

We construct a comprehensive feature representation capturing multiple aspects of each training sample, including task difficulty, lexical and syntactic complexity, readability, and semantic content. These features are designed to provide a rich, multi-dimensional characterization of the data, enabling effective clustering and curriculum design.

**Task Difficulty** To estimate the inherent difficulty of a sample, we compute its empirical accuracy based on $N$ repeated predictions from the same model: $f_{acc} = \frac{1}{N}\sum_{i=1}^{N}\mathbb{I}[\text{model}(x)_i = y^*]$, where $\mathbb{I}[\cdot]$ is the indicator function, and $\text{model}(x)_i$ denotes the $i$-th prediction made by the model on input $x$. A lower $f_{acc}$ indicates that the sample is consistently challenging for the model, while a higher value suggests that the sample is easier or already well-learned.

**Lexical and Syntactic Features** Lexical and syntactic complexity provide insights into the surface-level structure of the text. We extract word- and sentence-level statistics, including total word count $n_w$, average word length $\bar{l}_w$, sentence count $n_s$, average sentence length $\bar{l}_s$, and character composition (letters $n_{alpha}$, digits $n_{digit}$, symbols $n_{sym}$). These are combined into a lexical feature vector: $f_{lex} = [n_w, \bar{l}_w, n_s, \bar{l}_s, n_{alpha}, n_{digit}, n_{sym}]$. Longer or more complex sentences may increase cognitive load for the model.

**Readability Metrics** To assess how easily a text can be understood, we incorporate established readability metrics, including the Flesch Reading Ease (FRE) score: $f_{FRE} = 206.835 - 1.015\frac{n_w}{n_s} - 84.6\frac{n_{syl}}{n_w}$, where $n_{syl}$ is the syllable count, and the Gunning Fog Index. Samples with low readability scores are likely more linguistically complex, potentially requiring different learning strategies or longer attention spans during training.

**Semantic Features** To capture deeper conceptual and contextual information beyond surface text, we use the BAAI/bge-m3 multilingual embedding model Chen et al. (2024) to extract 1024-dimensional semantic vectors. These vectors encode high-level meaning and relationships within

the text, enabling clustering that reflects conceptual similarity rather than mere lexical similarity. We reduce the embedding dimensionality to 128 via PCA while retaining 95% of variance: $f_{sem}^{reduced} = \text{PCA}(\text{BGE-M3}(x), k = 128)$.

**Final Feature** The final representation concatenates all: $f = \text{Concat}[f_{acc}, f_{lex}, f_{FRE}, f_{sem}^{reduced}]$. We then apply z-score normalization to standardize each dimension: $\tilde{f}_i = \frac{f_i - \mu_i}{\sigma_i}$, where $\mu_i$ and $\sigma_i$ are the mean and standard deviation of the $i$-th feature across all samples, respectively.

**Clustering** We organize the training data into meaningful clusters using K-means++ Arthur & Vassilvitskii (2006); Pedregosa et al. (2011), where each sample is represented by its feature vector $f_i$ (as defined in 3.2). Let the set of clusters be $\mathcal{D} = \{d_1, \ldots, d_N\}$, and let $\mu_j$ denote the centroid of cluster $d_j$. The clustering objective minimizes the within-cluster variance: $\mathcal{L}_{cluster} = \sum_{j=1}^{N} \sum_{f_i \in d_j} \|f_i - \mu_j\|_2^2$. These clusters $\mathcal{D}$ define the distributions used in our curriculum learning, where each $d_j$ groups samples with similar difficulty and semantic properties, enabling adaptive, diversity-aware sampling in RLHF training. The necessity of fusing semantic and structural features will be discussed in Appendix B.

### 3.3 INFLUENCE FUNCTION COMPUTATION

To guide adaptive curriculum learning in RLHF, we extend influence functions to evaluate the training value of each data distribution. Let $\mathcal{D} = \{d_1, \ldots, d_N\}$ denote the set of clusters defined in the previous section, and let a sample $(x, o)$ be drawn from distribution $d_j$, with output $o \sim \pi_\theta(\cdot|x)$, where $\pi_\theta$ represents the policy of the language model parameterized by $\theta$. The influence of this sample on the validation performance is approximated by:

$$\mathcal{I}(x, o) = -\nabla_\theta \ell(x, o; \theta)^\top H_\theta^{-1} \nabla_\theta \mathcal{L}_{val}(\theta). \tag{1}$$

Here, $\ell(x, o; \theta)$ is the RLHF loss for the sample (e.g., GRPO loss), $H_\theta = \mathbb{E}_{(x,o) \in \mathcal{B}_{train}}[\nabla_\theta^2 \ell(x, o; \theta)]$ is the Hessian of the training loss, and $\mathcal{L}_{val}(\theta)$ is the validation loss. This standard formulation provides a principled estimate of how removing or weighting a sample affects overall model alignment.

Direct computation of $H_\theta^{-1}$ is intractable for LLMs. We adopt the Conjugate Gradient (CG) method to approximate $H_\theta^{-1}$ efficiently. We further introduce **distribution-level influence aggregation and sliding-window adaptation**:

- **Cluster-Level Aggregation:** Instead of computing influence per individual sample across the entire dataset, we aggregate influence scores within each cluster $d_j$ to obtain a cluster-level estimate: $\mathcal{I}_{d_j} = \frac{1}{|d_j|} \sum_{(x,o) \in d_j} |\mathcal{I}(x, o)|$. This reduces computation and emphasizes the collective value of semantically and structurally coherent samples.

- **Sliding Window for Dynamic Adaptation:** To reflect the evolving learning state of the model, we maintain a sliding window of recent influence scores $\mathcal{I}_{d_j}^w$ for each cluster:

$$\mathcal{I}_{d_j}^w \leftarrow \text{last}_k \left( \mathcal{I}_{d_j}^w \cup \{\mathcal{I}(x, o) \mid (x, o) \in \mathcal{B}_{t, d_j}\} \right), \tag{2}$$

where $\mathcal{B}_{t, d_j}$ is the subset of the batch from cluster $d_j$, and $k$ is the window size. This mechanism ensures that the sampling statistics reflect the model's current learning state rather than being dominated by early training dynamics, enabling subsequent adaptive selection strategies to prioritize distributions effectively.

Note that the Hessian-vector products $H_\theta p$ are computed via automatic differentiation, yielding $O(|\theta|)$ memory and $O(|\mathcal{B}| \cdot |\theta|)$ computation per CG iteration. In practice, we only compute influences for sampled batches, and CG typically converges within 10 iterations. Combined with cluster-level aggregation, this makes influence-guided sampling feasible at the LLM scale while preserving accuracy for curriculum learning decisions.

### 3.4 UCB-BASED DISTRIBUTION SELECTION

To dynamically prioritize high-value distributions while maintaining exploration, we frame the distribution selection process as a Multi-Armed Bandit (MAB) problem. Each distribution $d_j$ is treated

as an arm in the bandit setting, where pulling an arm corresponds to sampling from that distribution and receiving a reward based on its influence score. To balance exploration and exploitation, we adopt the Upper Confidence Bound (UCB) strategy, which selects the distribution with the highest optimistic estimate of its expected reward. For each distribution $d_j$, let $\mathcal{I}_{d_j}^w$ denote the sliding window of recent influence scores. We compute the Cluster Score as:

$$\text{CS}(d_j) = \hat{L}(d_j) + \sqrt{\frac{2\log(n_{\text{total}} + 1)}{n_{d_j} + 1}}, \tag{3}$$

where $\hat{L}(d_j) = \frac{1}{|\mathcal{I}_{d_j}^w|} \sum_{i \in \mathcal{I}_{d_j}^w} i$ is the mean absolute influence score within the sliding window, $n_{d_j}$ is the total number of samples seen from $d_j$, and $n_{\text{total}} = \sum_{j=1}^N n_{d_j}$. The exploitation term $\hat{L}(d_j)$ captures the expected learning value, while the exploration term encourages sampling of less-visited distributions. This formulation naturally increases the score for distributions that have been sampled less frequently, encouraging balanced exploration. The sampling probability for each distribution is computed using a softmax transformation with temperature parameter $\tau$:

$$P(d_j) = \frac{\exp(\text{CS}(d_j)/\tau_t)}{\sum_{j'=1}^N \exp(\text{CS}(d_{j'})/\tau_t)}, \tag{4}$$

with an adaptive temperature schedule $\tau_t = \tau_0 \cdot (1 - \gamma \cdot t/T)$, where $\tau_0$ is the initial temperature, $\gamma \in [0,1]$ is the decay rate, and $T$ is the total number of training steps. Early in training, this encourages broad exploration, gradually shifting toward exploitation of high-value distributions as learning progresses. The temperature parameter $\tau$ modulates the exploration-exploitation trade-off: lower values yield more deterministic selection favoring high-scoring arms, while higher values promote uniform exploration across all arms.

## 3.5 Integration with RLHF Training

We integrate our curriculum learning framework with GRPO as the underlying RLHF algorithm. For a batch $\mathcal{B}_t$ sampled according to the current distribution weights $P(d_j)$, the GRPO loss is defined as: $\mathcal{L}_{\text{GRPO}}(\mathcal{B}_t) = -\mathbb{E}_{(x,o) \in \mathcal{B}_t}\left[\frac{\pi_\theta(o|x)}{\pi_{\text{ref}}(o|x)} \cdot A(x,o)\right]$, where the advantage function $A(x,o) = r(x,o) - \bar{r}(\mathcal{B}_t)$ employs group normalization within the batch, and $\pi_{\text{ref}}$ is a reference policy to constrain updates and prevent excessive deviation from the pre-trained model behavior. The training process alternates between curriculum adaptation and policy optimization. After processing each batch through GRPO, we periodically recompute influence scores every $m$ steps to track the evolving learning dynamics. This periodic update strategy balances computational efficiency with adaptive accuracy, as influence patterns typically change gradually during training. The updated scores feed back into the UCB mechanism (Equation 3), which adjusts distribution probabilities for subsequent sampling, as summarized by the full pseudocode in Appendix A.

A key design choice is the temperature-controlled exploration schedule, which gradually transitions from broad exploration across all distributions to focused exploitation of high-value clusters. This annealing process ensures that early training benefits from diverse data exposure, while later stages concentrate on the most impactful samples for model refinement. The sliding window mechanism maintains temporal awareness, preventing the curriculum from being anchored to outdated learning statistics and enabling responsive adaptation to the model's current needs.

## 3.6 Computational Complexity Analysis

The primary computational cost of our framework arises from two components: influence function evaluation and UCB-based distribution updates. Let $N$ denote the number of distributions, $|\theta|$ the number of model parameters, $|\mathcal{B}|$ the batch size, $k$ the sliding window size, and $k_{\text{CG}}$ the number of conjugate gradient iterations for influence computation.

**Influence Function Evaluation** For a batch of size $|\mathcal{B}|$, computing influence functions via CG requires $O(k_{\text{CG}} \cdot |\mathcal{B}| \cdot |\theta|)$ operations, dominated by Hessian-vector products. Importantly, we only compute influences for sampled batches rather than the entire dataset, greatly reducing overhead.

**UCB Score Updates and Sampling** Updating sliding-window statistics and UCB scores across $N$ distributions incurs $O(N \cdot k)$ operations per step, while sampling according to the softmax distribution requires only $O(N)$ operations.

Thus, the overall per-step computational complexity is $O\big(k_{\mathrm{CG}} \cdot |\mathcal{B}| \cdot |\theta| + N \cdot k\big)$. In practice, the selective evaluation of influence functions and distribution-level curriculum learning improves training efficiency. Instead of computing influence scores at every step, they can be updated once every $m$ steps, where $m$ is a tunable parameter that balances accuracy and efficiency, yielding net computational savings while guiding the model toward high-value distributions effectively.

# 4 EXPERIMENTS

## 4.1 EXPERIMENTAL SETUP

**Datasets** We conduct comprehensive experiments on four mathematical reasoning benchmarks that span different difficulty levels and problem types. GSM8K Cobbe et al. (2021) contains 8,792 grade school math problems with natural language descriptions, providing a foundation for evaluating basic arithmetic and reasoning capabilities. The MATH dataset Hendrycks et al. (2021) presents 7,500 competition-level mathematics problems across seven subjects, including algebra, geometry, and number theory, offering a more challenging testbed for advanced mathematical reasoning. To assess performance on the most demanding problems, we include AIME24 Di Zhang (2025), consisting of 30 problems from the 2024 American Invitational Mathematics Examination, which requires sophisticated multi-step reasoning and creative problem-solving strategies. Additionally, we evaluate on MATH500 HuggingFaceH4 (2025), a carefully curated subset of 500 challenging problems from the MATH benchmark that OpenAI created Lightman et al. (2023), selected to represent diverse problem types while maintaining high difficulty.

**Baseline Methods** We compare our approach against three representative baseline methods. Random Sampling serves as the fundamental baseline, uniformly selecting samples from the entire dataset without considering any quality or difficulty metrics. DUMP Wang et al. (2025) employs a curriculum learning framework for RL-based LLM post-training, dynamically adjusting sampling probabilities across diverse data distributions based on policy advantages. AdaRFT Shi et al. (2025) uses adaptive curriculum learning, dynamically adjusting the difficulty of training problems based on recent reward signals in reinforcement finetuning. This method adapts to the model's evolving capabilities but relies solely on difficulty metrics without considering the actual influence of samples on learning progress.

**Implementation Details** All experiments are conducted using three representative language models of varying scales: Llama3.2-1B-Instruct, Llama3.2-3B-Instruct, and Qwen2.5-1.5B-Instruct. We implement GRPO training with a learning rate of $1 \times 10^{-6}$, batch size of 8, and train for 1 epoch on each dataset. The influence function computation uses Conjugate Gradient approximation with a tolerance of $10^{-6}$ and maximum iterations of 10. For the UCB bandit algorithm, we set the exploration parameter $c = 1.0$ and update cluster scores every 5 training steps. All experiments are conducted on 8 NVIDIA A100 GPUs with mixed precision training and repeated 3 times. More details can be found in C.

**Evaluation Metrics** We evaluate model performance using accuracy as the primary metric, calculated as the percentage of problems correctly solved with an exact match on final answers. For mathematical problems requiring numerical answers, we allow for minor formatting variations while ensuring mathematical equivalence.

## 4.2 MAIN RESULTS

**Overall Performance Comparison** Table 1 presents comprehensive performance comparisons across all evaluated models and datasets. Our method demonstrates consistent superiority, achieving the highest average accuracy across benchmarks for all three model architectures. On Llama3.2-1B-Instruct, we observe an average improvement of 1.681% over the strongest baseline, with particularly gains on GSM8K (60.071% vs. 59.540% for DUMP) and MATH500 (29.333% vs. 27.333% for AdaRFT). The performance advantage becomes more pronounced with larger models, as ev-

Table 1: Test Accuracy of Different Methods on Mathematical Benchmarks

| Model | Method | GSM8K | MATH | AIME24 | MATH500 | Average |
|-------|--------|-------|------|--------|---------|---------|
| Llama3.2-1B-Instruct | Random | 58.453 | 4.1683 | 6.4171 | 24.667 | 23.426 |
| | AdaRFT | 55.421 | **10.091** | 4.6346 | 27.333 | 24.370 |
| | DUMP | 59.540 | 4.8686 | 4.6346 | 27.333 | 24.094 |
| | Ours | **60.071** | 8.2033 | **6.5954** | **29.333** | **26.050** |
| Llama3.2-3B-Instruct | Random | 81.927 | 16.489 | 24.066 | 45.333 | 41.954 |
| | AdaRFT | 81.072 | 13.185 | 15.686 | 45.333 | 38.819 |
| | DUMP | 82.739 | 19.081 | 21.747 | **49.667** | 43.309 |
| | Ours | **83.422** | **19.361** | 24.599 | 46.667 | **43.512** |
| Qwen2.5-1.5B-Instruct | Random | 78.115 | 5.5852 | 5.7041 | 48.000 | 34.351 |
| | AdaRFT | 75.663 | 10.258 | 5.3476 | 49.000 | 35.067 |
| | DUMP | **78.241** | 5.8968 | 5.3476 | **52.333** | 35.455 |
| | Ours | 78.140 | **11.682** | 5.8824 | 52.333 | **37.009** |

idenced by Llama3.2-3B-Instruct achieving 43.512% average accuracy compared to 41.454% for Random sampling.

The results reveal interesting patterns in how different methods perform across varying problem difficulties. While AdaRFT shows competitive performance on moderately difficult problems, our influence-based selection excels on both extremes—basic problems in GSM8K and highly challenging problems in AIME24. This suggests that our method effectively identifies training samples that provide maximal learning signal regardless of inherent difficulty, rather than relying on predefined difficulty estimates or confidence metrics alone.

Particularly striking are the results on AIME24, where our method achieves 24.599% accuracy on Llama3.2-3B compared to 15.686% for AdaRFT, representing a relative improvement of 56.8%. This substantial gain on competition-level problems indicates that influence-based selection is especially valuable when dealing with complex, multi-step reasoning tasks where traditional difficulty metrics may fail to capture the true learning value of samples.

**Training Efficiency Analysis** Beyond final performance improvements, our method demonstrates superior training efficiency across all evaluated scenarios. Figure 3 illustrates the learning dynamics on the AIME24 and GSM8K, revealing several key advantages of our approach. Our method enables models to achieve higher performance with reduced variance, translating to faster convergence in training scenarios.

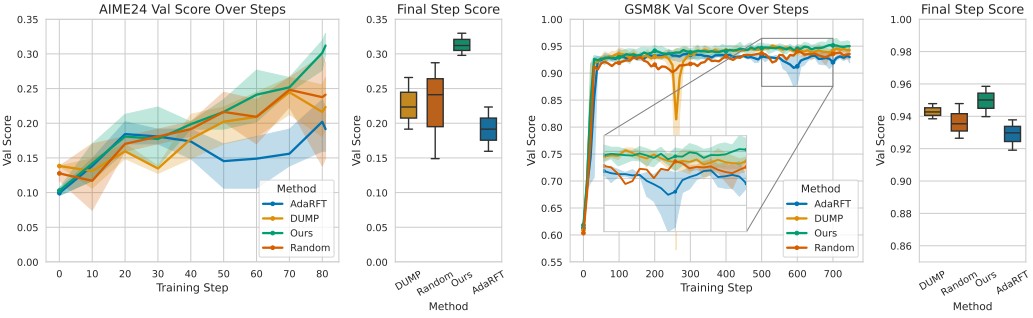

Figure 3: Validating curves of different methods. Our approach (green) demonstrates faster convergence and reduced variance compared to baseline methods, achieving higher final performance.

The training curves reveal that our method maintains more stable learning progress throughout the training process, with significantly reduced variance compared to random sampling approaches. While baseline methods often exhibit plateau periods where learning stagnates, our influence-based selection consistently provides informative samples that drive continued improvement. This stability stems from the principled approach to identifying high-impact training instances rather than relying on heuristic selection strategies.

Table 2: Ablation study results, IF=influence function (Ours), ADV=advantage, PPL=perplexity

| Model | Method | GSM8K | MATH | AIME24 | MATH500 | Average |
|---|---|---|---|---|---|---|
| **Llama3.2-1B-Instruct** | Bandit+ADV | 59.540 | 4.868 | 4.634 | 27.333 | 24.094 |
| | Bandit+PPL | **60.955** | 4.281 | 5.347 | 24.000 | 23.646 |
| | Bandit+IF | 60.071 | **8.203** | **6.595** | **29.333** | **26.051** |
| **Llama3.2-3B-Instruct** | Bandit+ADV | 82.739 | 19.081 | 21.747 | **49.667** | 43.309 |
| | Bandit+PPL | 83.205 | **19.661** | 23.529 | 46.071 | 43.117 |
| | Bandit+IF | **83.422** | 19.361 | **24.599** | 46.667 | **43.512** |
| **Qwen2.5-1.5B-Instruct** | Bandit+ADV | 78.241 | 5.896 | 5.347 | 52.333 | 35.455 |
| | Bandit+PPL | 76.876 | 8.433 | 5.417 | 51.000 | 35.432 |
| | Bandit+IF | 78.140 | **11.682** | **5.882** | 52.333 | **37.009** |

Most notably, our method successfully balances exploitation of valuable data clusters with exploration of undersampled regions, preventing the model from overfitting to specific problem patterns. The dynamic nature of our approach allows it to adapt sample selection as the model's capabilities evolve, ensuring that training resources are allocated optimally throughout the learning process.

### 4.3 ABLATION STUDIES

To dissect the contributions of individual components in our framework, we conduct systematic ablation studies comparing our full method against variants that use alternative selection criteria. The results, presented in Table 2, demonstrate the importance of influence-based scoring in achieving optimal performance.

The ablation analysis reveals that replacing influence functions with simple advantage-based selection (ADV) leads to performance degradation across most benchmarks, particularly on challenging datasets like MATH and AIME24. This confirms our hypothesis that advantage scores alone, while useful for immediate reward estimation, fail to capture the long-term learning impact of training samples. The influence function component provides crucial insight into how individual samples affect the model's overall capability development.

Substituting influence scores with perplexity-based selection (PPL) yields mixed results, performing reasonably well on some datasets but failing to achieve the consistent improvements seen with our full method. Perplexity-based selection tends to favor samples that are moderately difficult for the current model state, but may miss samples that, while appearing easy or difficult, provide crucial learning signals for capability generalization.

The comparison between our full method and these ablated versions highlights the synergistic effect of combining influence-based quality assessment with bandit-driven exploration. Neither component alone achieves the robust performance improvements observed when they work in conjunction, suggesting that both principled sample valuation and adaptive distribution balancing are essential for effective curriculum learning in RL-based post-training.

## 5 CONCLUSION

This paper presents an influence-guided curriculum learning framework for RL-based post-training of large language models. Our method leverages influence functions to quantify the training value of data distributions and dynamically adjusts sampling strategies throughout the learning process. By organizing training data into clusters and treating each cluster as an arm in a multi-armed bandit framework, we enable principled balancing between exploitation of high-impact distributions and exploration of underrepresented regions. The integration of sliding-window influence computation ensures that our curriculum adapts to the model's evolving learning state, while the UCB-based scheduler provides theoretical guarantees for optimal distribution selection. Extensive experiments on mathematical reasoning benchmarks demonstrate consistent improvements over existing methods, with particularly substantial gains on challenging competition-level problems, highlighting the importance of principled, adaptive curriculum design in maximizing the effectiveness of RL-based post-training for complex reasoning tasks.

## REPRODUCIBILITY STATEMENT

We have made efforts to ensure the reproducibility of our results. While Section 4.1 introduces the datasets used, the detailed data preprocessing steps are provided in the submitted source code. The experimental environment and key hyperparameters are also described there, and further hyperparameter configurations are listed in Appendix C. To facilitate replication, we have included an anonymous link https://anonymous.4open.science/r/InfluenceCB/ to the complete source code and data in the supplementary materials. The repository contains all necessary scripts for data processing, model training, and evaluation, enabling reproduction of our results.

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

## A  PSEUDO CODE OF THE PROPOSED METHOD

---

**Algorithm 1** Automated Distribution-Level Curriculum Learning with UCB Sampling

---

1: **Input:** Dataset $\mathcal{D} = \{d_1, \ldots, d_N\}$; pre-trained model parameters $\theta$
2: **Output:** Post-trained model parameters $\theta$
3: ▷ Initialize distribution-level statistics
4: **for all** $d_j \in \mathcal{D}$ **do**
5:     $\mathcal{I}_{d_j}^w \leftarrow []$                                      ▷ Sliding window for influence scores
6:     $n_{d_j} \leftarrow 0$                                       ▷ Total samples seen from $d_j$
7:     $P(d_j) \leftarrow \frac{1}{N}$                                   ▷ Equal initial weights
8: **end for**
9: **for** training step $t = 1, 2, \ldots, T$ **do**
10:     Sample batch $\mathcal{B}_t$ from $\mathcal{D}$ according to $P(d_j)$
11:     Compute influence functions $\mathcal{I}(o)$ for all $o \in \mathcal{B}_t$ using the CG method
12:     **for all** $d_j$ with samples in $\mathcal{B}_t$ **do**
13:         $n_{d_j} \leftarrow n_{d_j} + |\mathcal{B}_{t,d_j}|$              ▷ $\mathcal{B}_{t,d_j}$: subset of batch from $d_j$
14:         $\mathcal{I}_{d_j}^w \leftarrow \mathcal{I}_{d_j}^w \cup \{ |\mathcal{I}(o)| \mid x \in \mathcal{B}_{t,d_j},\ o \sim \pi_\theta(\cdot \mid x) \}$     ▷ Append new influence scores
15:         $\mathcal{I}_{d_j}^w \leftarrow \mathrm{last}_k(\mathcal{I}_{d_j}^w)$                    ▷ Keep last $k$ elements
16:     **end for**
17:     ▷ Compute cluster scores for each distribution
18:     $n_{\text{total}} \leftarrow \sum_{d_j \in \mathcal{D}} n_{d_j}$
19:     **for all** $d_j \in \mathcal{D}$ **do**
20:         $\hat{L}(d_j) \leftarrow \frac{1}{|\mathcal{I}_{d_j}^w|} \sum_{i \in \mathcal{I}_{d_j}^w} i$          ▷ Mean of absolute influence scores
21:         $\mathrm{CS}(d_j) \leftarrow \hat{L}(d_j) + \sqrt{\frac{2 \log(n_{\text{total}}+1)}{n_{d_j}+1}}$     ▷ Update sampling distribution
22:         $P(d_j) \leftarrow \frac{\exp(\mathrm{CS}(d_j)/\tau)}{\sum_{j'=1}^{N} \exp(\mathrm{CS}(d_{j'})/\tau)}$          ▷ $\tau$: temperature
23:     **end for**
24:     Update $\theta$ using $\mathcal{B}_t$ with an RL algorithm (e.g., GRPO)
25: **end for**
26: **return** $\theta$

---

## B  CLUSTERING ANALYSIS

This section demonstrates the critical necessity of our proposed multi-faceted feature representation by contrasting two unimodal clustering approaches visualized in the provided figure. The left plot of Figure 4 illustrates clustering based purely on semantic embeddings, which captures topical similarity, while the right plot shows clustering derived from lexical and syntactic features, which serve as a proxy for structural complexity and difficulty. Together, they reveal the inherent limitations of relying on any single feature dimension and underscore the imperative for a hybrid approach to construct meaningful data distributions for curriculum learning.

Clustering in the semantic embedding space, as shown in the left of Figure 4, successfully groups samples by conceptual similarity. However, this method entirely disregards structural attributes. The resulting clusters are highly heterogeneous in terms of text length—a key indicator of complexity—as evidenced by the indiscriminate mixture of colors (from yellow for short texts to red for long texts) within single cluster boundaries. Consequently, a simple definition and a complex, multi-step application problem concerning the same topic could be placed in the same cluster. For our curriculum learning framework, such clusters are ineffective; they provide the Multi-Armed Bandit (MAB) scheduler with ambiguous and unpredictable arms, hindering its ability to learn an optimal sampling policy based on task difficulty.

Conversely, the right plot of Figure 4, which utilizes lexical and syntactic features, excels at organizing the data by structural complexity. It produces a highly ordered manifold where text length varies smoothly across the distribution, creating clusters that are remarkably homogeneous in terms of structural attributes. Yet, this approach completely ignores the underlying semantic content. Sam-

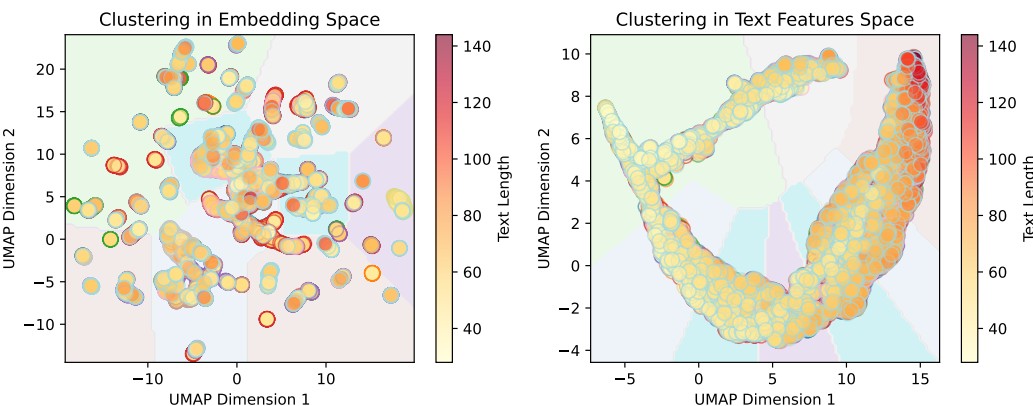

Figure 4: The figure shows the clustering results of the GSM8K dataset in two different feature spaces, and visualizes the results using Unified Mapping (UMAP) dimensionality reduction. The color of the data points represents the text length, a key vocabulary and difficulty feature.

ples with similar word counts and sentence structures, such as a mathematics problem and a code generation snippet, could be erroneously grouped together despite requiring fundamentally different reasoning capabilities from the model. This lack of semantic coherence renders the clusters equally problematic, as the MAB scheduler can discern the difficulty of a cluster but remains ignorant of its topic or knowledge domain.

These two visualizations represent two extremes, each capturing a vital but incomplete aspect of the data. The semantic approach understands *what* a sample is about but not *how hard* it is, while the structural approach understands *how hard* it is but not *what* it is about. An ideal cluster for an adaptive curriculum must be homogeneous along both axes, representing a coherent and well-defined learning objective, such as "simple mathematical concepts" or "complex logical reasoning tasks." Our proposed method, which fuses semantic, lexical, difficulty, and readability features into a single comprehensive vector, is designed precisely to achieve this synthesis. By creating clusters that are consistent in both content and complexity, we provide the MAB scheduler with meaningful, interpretable arms, enabling it to make intelligent, adaptive decisions. This foundation is a prerequisite for accelerating convergence and enhancing the final performance of the model, thereby validating that a fused feature representation is not merely beneficial but essential for effective curriculum learning in RLHF.

## C  HYPERPARAMETERS

We use Verl Sheng et al. (2024) as a code-base to conduct all experiments; the hyperparameters are listed in Table 3.

## D  THE USE OF LARGE LANGUAGE MODELS

Large Language Models were employed as general-purpose assistive tools throughout the research process. Specifically, LLMs were used to aid and polish the writing of this manuscript, including refining grammar, improving clarity, and restructuring sentences for better readability. Additionally, LLMs facilitated literature retrieval and discovery by suggesting relevant prior work based on topic descriptions and keywords, which helped broaden the scope of our references.

Beyond writing and literature support, LLMs contributed to the development of dataset processing scripts and preliminary data analysis code. These included generating boilerplate code for data cleaning, formatting, and visualization, which were subsequently reviewed and modified by the authors to ensure correctness and alignment with research goals.

Table 3: Hyperparameters for GRPO Training

| Hyperparameter | Value |
|---|---|
| Max Prompt Length | 1024 |
| Max Response Length | 3072 |
| Use Dynamic Batch Size | True |
| Actor PPO Max Token Length | 9500 |
| Infer PPO Max Token Length | 9500 |
| Offload | False |
| Generation Tensor Parallelism (gen_tp) | 1 |
| Train Batch Size | 8 |
| Algorithm Adv Estimator | grpo |
| Data Train Batch Size | 8 |
| Data Enable Curriculum Learning | True |
| Data Shuffle | True |
| Data Influence Cluster Sampling Interval | 5 |
| Data Max Prompt Length | 1024 |
| Data Max Response Length | 3072 |
| Actor Rollout Ref Actor Use Dynamic Bsz | True |
| Actor Rollout Ref Ref Log Prob Use Dynamic Bsz | True |
| Actor Rollout Ref Rollout Log Prob Use Dynamic Bsz | True |
| Actor Rollout Ref Actor PPO Max Token Len Per GPU | 9500 |
| Actor Rollout Ref Ref Log Prob Max Token Len Per GPU | 9500 |
| Actor Rollout Ref Rollout Log Prob Max Token Len Per GPU | 9500 |
| Actor Rollout Ref Actor Ulysses Sequence Parallel Size | 1 |
| Actor Rollout Ref Ref Ulysses Sequence Parallel Size | 1 |
| Actor Rollout Ref Actor Optim LR | 1e-6 |
| Actor Rollout Ref Model Use Remove Padding | True |
| Actor Rollout Ref Actor PPO Mini Batch Size | 32 |
| Actor Rollout Ref Actor Use KL Loss | True |
| Actor Rollout Ref Actor KL Loss Coef | 0.001 |
| Actor Rollout Ref Actor KL Loss Type | low_var_kl |
| Actor Rollout Ref Model Enable Gradient Checkpointing | True |
| Actor Rollout Ref Actor FSDP Config Param Offload | False |
| Actor Rollout Ref Actor FSDP Config Optimizer Offload | False |
| Actor Rollout Ref Rollout Name | vllm |
| Actor Rollout Ref Rollout GPU Memory Utilization | 0.6 |
| Actor Rollout Ref Rollout N | 6 |
| Actor Rollout Ref Rollout Enforce Eager | True |
| Actor Rollout Ref Rollout Free Cache Engine | False |
| Actor Rollout Ref Rollout Enable Chunked Prefill | True |
| Actor Rollout Ref Rollout Tensor Model Parallel Size | 1 |
| Actor Rollout Ref Ref FSDP Config Param Offload | False |
| Algorithm KL Ctrl KL Coef | 0.001 |
| Trainer Critic Warmup | 0 |
| Trainer N GPUs Per Node | 8 |
| Trainer Nnodes | 1 |
| Trainer Save Freq | 200 |
| Trainer Test Freq | 10 |
| Trainer Total Epochs | 1 |

All outputs from LLMs were critically evaluated and edited by the authors, and no content was used without verification. The use of LLMs did not replace human intellectual contributions but served to accelerate and enhance various stages of the research workflow.

