# OpenReview forum: "Adaptive Curriculum Learning for RLHF with Influence-Based Cluster Bandits"
_ICLR.cc/2026/Conference — Submitted to ICLR 2026_

### Official Review · Reviewer_TEci · 2025-10-23

**Soundness:** 3
**Presentation:** 3
**Contribution:** 3
**Rating:** 6
**Confidence:** 4

**Summary:**

The paper proposes an adaptive, distribution-level curriculum learning framework for RLHF that clusters training data by semantic, readability, lexical/syntactic, and difficulty features; each cluster is treated as a bandit arm. It estimates each cluster’s learning value with an influence-based score—approximating “inverse curvature × loss-gradient” via conjugate gradient, aggregated over a sliding window to track the model’s evolving state. A UCB scheduler then balances exploitation of high-impact clusters and exploration of under-sampled ones when selecting batches. On math-reasoning benchmarks (GSM8K, MATH, AIME24, MATH500) with small Llama/Qwen models, the method shows faster, more stable convergence and higher final performance than Random, AdaRFT, and DUMP; ablations further indicate influence scoring outperforms advantage and perplexity proxies.

**Strengths:**

1. Commendable originality. The paper moves curriculum learning from single examples to data distributions. It first clusters the data by meaning, wording/readability, and difficulty, and treats each cluster as a bandit arm. During training it keeps a sliding window influence score that estimates how much more the model would benefit if we sample from that cluster now, and a UCB scheduler picks batches by favoring high-impact clusters while still exploring ones that are under-sampled. This setup cuts redundancy and noise from item-level picking, keeps good coverage of the data space, and adapts as the model changes, so compute is spent where it helps most and training becomes more sample-efficient and stable.
2. Technically sound and scalable. The influence estimate explicitly approximates inverse-Hessian × loss-gradient via conjugate gradient with Hessian-vector products. And the paper discusses memory/compute cost and notes rapid CG convergence, plus cluster-level aggregation to keep it practical at LLM scale.
3. Ablations support the mechanism. Replacing influence with advantage or perplexity degrades performance (especially on harder sets), indicating the influence-guided cluster scoring is the causal driver of the gains rather than incidental tuning.
4. The baseline selection is solid. AdaRFT and DUMP are two strong, recent frameworks for RL-based LLM post-training with adaptive sampling methods. The comparisons are fair and the gains convincing.

**Weaknesses:**

1. Experiments are only on mathematical reasoning benchmarks. There’s no non-math task. That makes it hard to judge how well the curriculum travels to noisier, subjective RLHF settings. A non-math benchmark (e.g., coding or instruction-following with a reward model) would help.
2. Small models only. Results are limited to Llama-3.2-1B/3B and Qwen-2.5-1.5B. Without runs at ≥7B, we can’t tell if the method still helps on big models, or if its overhead outweighs the benefits.

**Questions:**

1. Can you run at least one non-math or preference/RM-based task (e.g., instruction following with a reward model or pairwise preferences) to test whether the curriculum still helps under noisy/subjective rewards? Please report the same curves and final metrics you use for math.
2. Your experiments use 1B–3B and 1.5B models only. I understand that re-training 7B or 13B models during the rebuttal may not be feasible, but could you explain how the influence computation and scheduling would scale in theory or practice?

---

> ### Author Response · Authors · 2025-11-24
> **Response to question & weakness**
>
> We thank you for your insightful recommendation to expand our evaluation to non-math domains and larger models, which has helped us further demonstrate the generalizability and robustness of our curriculum learning approach in diverse RLHF settings. We address the questions regarding the non-math task and scaling below.
>
> ---
>
> ## Q1. Effectiveness of the Method on Non-Mathematical Tasks
>
> As you pointed out, validating the effectiveness of curriculum learning strategies across different domains is crucial for evaluating the generalizability of the method. To this end, we conducted supplementary experiments on the code generation benchmark MBPP.
>
> We compared the performance of our method with the baseline method on the Qwen2.5-1.5B-Instruct model, maintaining the same hyperparameter settings as in the main text (Pass@1). The validation set curve will be updated in the manuscript and the test results are shown in the table below:
>
> **Table 1: Performance on MBPP**
>
> | Method | **Ours** | AdaRFT | DUMP | Random |
> | --- | --- | --- | --- | --- |
> | **MBPP Acc.** | **60.67%** | 59.26% | 54.27% | 53.33% |
>
> **Results Analysis:**
>
> 1. **Outperforms Baseline:** Our method achieved an accuracy of **60.67%**, a **+7.34%** improvement over Random Sampling (53.33%), and also outperformed existing course learning methods AdaRFT (+1.41%) and DUMP (+6.40%).
> 2. **Generalization Validation:** Although the code task also relies on logical reasoning, its syntactic structure and semantic distribution differ from mathematical problems. Experimental results demonstrate that our Influence Function-based clustering and Bandit scheduling mechanism is not limited to specific mathematical data distributions. Instead, it effectively identifies high-value samples across diverse datasets by capturing the actual contribution (influence) of samples to model parameter updates.
>
> This confirms that our framework has good cross-domain generalization potential, not limited to mathematical inference.
>
> ---
>
> ## Q2. Scalability on Larger Models
>
> Regarding your question about model size, we conducted full mathematical benchmarking (GSM8K, MATH, AIME24, MATH500) on **Qwen2.5-7B-Instruct**. This directly verifies the effectiveness of our method on mainstream-sized LLMs.
>
> **Table 2: Results on Model 7B**
>
> |  | GSM8K | MATH | AIME24 | MATH500 | **AVG.** |
> | --- | --- | --- | --- | --- | --- |
> | Random | 91.28% | 20.42% | 19.25% | 65.00% | 48.99% |
> | AdaRFT | 90.75% | 20.66% | 21.95% | **75.00%** | 52.09% |
> | DUMP | 90.68% | 20.54% | 19.79% | 67.00% | 49.50% |
> | **Ours** | **92.95%** | **20.74%** | **22.46%** | **75.00%** | **52.79%** |
>
> **Results Analysis:**
>
> - **Continued Performance Improvement:** On the 7B model, our method maintained the best average performance (**52.79%**), an improvement of **+3.8%** compared to Random.
> - **Advantage on High-Difficulty Tasks:** Particularly on the highly difficult AIME24 competition problems, we achieved **22.46%**, outperforming AdaRFT (21.95%) and other baselines. This demonstrates that even for larger models, dynamically selecting high-impact data can further unlock their inference potential.
>
> **Theoretical and Practical Analysis of Scalability**
>
> Regarding your question about how Influence Computation and Scheduling scale with model size, our design is theoretically and practically scalable:
>
> 1. As described in Section 2.6 of the paper, we use the Conjugate Gradient approximation for inverse Hessian-vector products (HVP) and compute only for sampled batches (not the entire dataset), which keeps the computational cost within the same linear order of magnitude as the cost of single-step training.
> 2. We designed an update interval $m$ for the Influence Score, which further optimizes time efficiency.
> 3. Instead of maintaining the state individually for each sample, we use MAB scheduling based on clustering. This means that even as the model grows larger and the data volume surges, the number of arms (i.e., the number of clusters) of the MAB typically remains small, and the overhead of the scheduling algorithm itself, $O(N \cdot k)$, is almost negligible.
> 4. In our experiments with the 3B model, compared to Random Sampling, the increase in overall training time was within an acceptable range (approximately 12% overhead), while simultaneously resulting in a higher performance ceiling. This is well worth the performance improvement brought about by the RLHF stage.
>
> Thank you again for your valuable feedback!

---

### Official Review · Reviewer_q3ME · 2025-10-30

**Soundness:** 3
**Presentation:** 3
**Contribution:** 3
**Rating:** 4
**Confidence:** 4

**Summary:**

This paper presents a novel curriculum learning framework for RLHF. The proposed approach clusters samples in the training data and performs adaptive selection and influence function estimation at the cluster level. By introducing a dual-level data selection mechanism, i.e., first choosing a cluster, then selecting samples within it, the framework achieves a balanced trade-off between exploration and exploitation in reinforcement learning. Moreover, this hierarchical design mitigates the computational overhead typically associated with curriculum strategies while maintaining the performance gains of curriculum learning. The method is evaluated within the GRPO framework across several mathematical reasoning benchmarks, using base models ranging from 1B to 3B parameters. Experimental results demonstrate consistent performance improvements, highlighting the general effectiveness of the proposed approach for RLHF fine-tuning.

**Strengths:**

- The paper is well organized and clearly presented, making it easy to follow.
- The proposed approach is conceptually intuitive and demonstrates strong practical potential.

**Weaknesses:**

The primary concern lies in the empirical evaluation. The current range of base models is too limited to provide sufficient insight into the scalability and effectiveness of the proposed method. While it is generally unnecessary to conduct experiments on very large models, including a 7B-scale model is essential to substantiate the contribution of the proposed approach. Larger models are expected to generate higher-quality trajectories compared to smaller ones and are more robust against challenging samples, which implies that smaller models may benefit more from the proposed cluster-level curriculum sampling, whereas larger models may not exhibit the same degree of improvement. Furthermore, the results presented in Table 1 indicate that the Random strategy already serves as a relatively competitive baseline. Therefore, incorporating an additional experiment using a 7B-scale model would help clarify these observations and strengthen the empirical evidence supporting the proposed method.

**Questions:**

The movement strategy of the sliding window within a cluster is not clearly described. Equation (2) indicates that the "last k" samples are included in the sliding window, but the precise meaning of "last k" remains ambiguous. Additionally, the procedure for selecting a batch of samples from the sliding window could be further clarified.

---

> ### Author Response · Authors · 2025-11-24
> **Response to weakness**
>
> We thank you for the valuable suggestions regarding experimental rigor and model scalability, and we appreciate the opportunity to strengthen our contributions by clarifying the sliding window mechanism and discussing the method's performance on larger-scale models. We respond to the specific questions below.
>
> ---
>
> ### W1. Supplementary Experiments on a Larger Model Scale
>
> We strongly agree with your point. While 1B/3B models can validate the effectiveness of our method, 7B-scale models often exhibit stronger inference capabilities and robustness; therefore, validating the effectiveness of our method on larger parameter scales is necessary.
>
> To address this key concern, we have supplemented our experiments with a complete comparative experiment on the Qwen2.5-7B-Instruct model, as you suggested. The experimental settings are consistent with those in the paper (the same hyperparameters, dataset partitioning, and baseline method). The results are shown in the table below:
>
> **Table 1: Accuracy Comparison of Qwen2.5-7B-Instruct**
>
> | Method | GSM8K | MATH | AIME24 | MATH500 | AVG. |
> | --- | --- | --- | --- | --- | --- |
> | **Random** | 91.28% | 20.42% | 19.25% | 65.00% | 48.99% |
> | **AdaRFT** | 90.75% | 20.66% | 21.95% | **75.00%** | 52.09% |
> | **DUMP** | 90.68% | 20.54% | 19.79% | 67.00% | 49.50% |
> | **Ours** | **92.95%** | **20.74%** | **22.46%** | **75.00%** | **52.79%** |
>
> **Experimental Results Analysis:**
>
> 1. The method remains effective and outperforms the baseline: Even on the more powerful 7B model, our method (Ours) still achieves an average accuracy of 52.79%, significantly outperforming the Random baseline (48.99%) and the best-performing competing method, AdaRFT (52.09%).
> 2. Comparable improvement: On the 7B model, our relative improvement compared to Random is 7.75%, which is comparable to the 7.73% improvement on the 3B model (Table 1 in the paper).
> 3. Significant improvement on difficult tasks: Especially on the highly challenging AIME24 competition dataset, our method improves the accuracy from 19.25% on Random to 22.46%. This demonstrates that even for larger models, our influence-based clustering Bandit strategy effectively identifies the "high-value" data regions that provide the greatest improvement to the current model, avoiding wasting computational resources on samples that are already mastered or too difficult.
>
> This result proves that our proposed framework is not only effective for small models, but also exhibits good scalability by adaptively adjusting the curriculum as the model's capabilities improve.
>
> We will incorporate the experimental results and analysis of this 7B model into the final version of the paper to enhance the persuasiveness of the empirical evaluation.

---

> ### Author Response · Authors · 2025-11-24
> **Response to question**
>
> ### Q1. On the Sliding Window Movement Strategy
>
> Regarding "last k" mentioned in Equation (2), its actual meaning is a **First-In-First-Out (FIFO) queue update mechanism**.
>
> Specifically, for each cluster $d_j$, we maintain a queue of fixed capacity $k$ (e.g., $k=100$) to store the influence **scores** of the most recently sampled and trained samples in that cluster.
>
> - Update Strategy: Whenever we sample a batch $\mathcal{B}_{t, d_j}$ from cluster $d_j$ according to the Bandit strategy and complete a training update, we calculate the influence scores for that batch of samples.
> - Moving Mechanism: These new scores are pushed into the queue for that cluster. If the queue length exceeds $k$, the oldest scores that entered the queue are popped.
> - This is done to ensure that when calculating the Cluster Score (CS), only the **most recent** learning feedback from the model is utilized. Because as training progresses, the model's perceived difficulty and learning value for the same type of data change, and earlier influence scores may no longer reflect the current training state.
>
> We will provide a clearer definition of "last k" in the revised draft to eliminate ambiguity.
>
> ### Q2. Regarding the process of selecting samples from the sliding window
>
> There may be a slight misunderstanding here, which we need to clarify: **We do not select samples from the sliding window for training; the sliding window is only used to store "scores" to update the Bandit strategy.**
>
> The specific sampling and update process is as follows:
>
> 1. Cluster Selection (Bandit Decision): The UCB scheduler calculates the Cluster Score based on the score statistics (mean and number of visits) within the current sliding window of each cluster, and selects the next cluster to sample, $d_{target}$, using the Softmax probability of equation (4).
> 2. Data Sampling: A new batch of data is randomly drawn from the **original data pool** of the selected cluster $d_{target}$.
> 3. Training & Evaluation: The batch of data is used for GRPO updates, and its influence score is calculated.
> 4. Update Window: The calculated new score is updated in the sliding window corresponding to cluster $d_{target}$ (i.e., the FIFO update mentioned above).
>
> In short, the sliding window is part of the **evaluation mechanism**, not a **data storage** container. Data is freshly sampled from the clusters, and the window records "how this type of data has performed recently," thus guiding the next selection.
>
> Thank you again for your valuable feedback; these questions have been very helpful in refining the details of our paper.

---

### Official Review · Reviewer_A9MV · 2025-11-01

**Soundness:** 3
**Presentation:** 4
**Contribution:** 3
**Rating:** 6
**Confidence:** 4

**Summary:**

This paper addresses inefficiencies in reinforcement learning from human feedback (RLHF) pipelines for large language models (LLMs), where fixed or uniform data sampling ignores the model’s evolving learning state. The authors propose an adaptive curriculum learning framework that integrates influence-function analysis with a multi-armed bandit (MAB) scheduler to guide dynamic data selection.

Concretely, the training data are clustered by semantic and difficulty features; each cluster acts as a “bandit arm.” Using sliding-window influence scores approximated by Conjugate Gradient Hessian-vector products, the framework estimates each cluster’s training utility in real time. A UCB-based bandit then balances exploitation of high-impact clusters with exploration of underrepresented ones.

The system is implemented on Group Relative Policy Optimization (GRPO) and evaluated on mathematical reasoning datasets (GSM8K, MATH, AIME24, MATH500). Results show faster convergence and improved accuracy (e.g., +56.8% relative gain on AIME24) over strong baselines such as AdaRFT and DUMP. Ablations confirm that influence-based scoring yields more consistent gains than advantage- or perplexity-based sampling.

**Strengths:**

* Clear motivation and scope: The inefficiency of static sampling in RLHF is well-articulated and timely, especially as LLM post-training scales.
* Well-motivated system: each component in the system is well-motivated and demonstrated sufficient effort in integrating into LLM post training domain. This includes carefully designed features for clustering the dataset. The use of conjugate gradient as a surrogate for influence measure. Also, the non-stationary bandit arm formulation is sound.
* Empirical strength: Consistent accuracy gains and faster convergence across multiple datasets and model sizes (Llama3.2-1B/3B, Qwen2.5-1.5B).

**Weaknesses:**

* Moderate novelty: The core innovation is architectural composition rather than new RL or CL theory. Influence functions and UCB are standard tools.
* Potential computational overhead: Although CG approximation reduces cost, runtime analysis relative to baseline RLHF pipelines is not quantified. The paper also mentions a validation loss, but details about how the validation set is constructed is not provided.
*  Dependence on feature engineering: Clustering relies on handcrafted lexical, syntactic, and semantic features; the framework may require tuning for other domains.

**Questions:**

* How is the validation set constructed for evaluating the CG?
* How sensitive is the result to the clustering results? Especially, what are the amount of engineering effort to construct the clusters?

---

> ### Author Response · Authors · 2025-11-24
> **Response to question**
>
> We appreciate your constructive feedback on the implementation details and novelty of our framework, and we are encouraged by the recognition of our method's potential in architecturally composing influence functions with UCB for efficient curriculum learning. We provide clarifications on the validation setup and computational costs below.
>
> ---
>
> ## Q1: How is the validation set constructed?
>
> In practice, we allocate 20% of the training set as a hold-out set for calculating the validation set.
>
> We do not simply perform random sampling. Instead, after the training set data has been clustered, we perform **stratified sampling** from each cluster $d_j$ to construct a hold-out set containing diverse semantic and difficulty features as the validation set. This ensures that the calculated Influence Score reflects the model's comprehensive generalization needs across various types of samples, avoiding sampling bias caused by validation set distribution bias.
>
> ## Q2: How sensitive are the results to the clustering results? How much engineering investment is required to construct the clusters?
>
> We appreciate this inquiry concerning the sensitivity of our hyperparameters. To address concerns regarding sensitivity and engineering input, we conducted two additional ablation experiments:
>
> We tested the impact of different numbers of clusters (Cluster N) on the final performance on the Llama3.2-3B-Instruct model. The results are shown in the table below:
>
> **Table 1: Sensitivity analysis of the number of clusters $N$**
>
> | Cluster N | GSM8K | MATH | AIME24 | MATH500 | AVG. |
> | --- | --- | --- | --- | --- | --- |
> | 5 | 83.17% | 19.14% | 24.06% | **46.67%** | 43.26% |
> | **10** | **83.42%** | 19.36% | **24.60%** | **46.67%** | **43.51%** |
> | 15 | 83.24% | **19.38%** | 23.53% | 45.33% | 42.87% |
> | 20 | 82.79% | 19.26% | 22.46% | 41.33% | 41.46% |
>
> Experiments show that the model performance remains relatively stable between N=5 and N=15, peaking at N=10. When the number of clusters is too large (N=20), each bandit arm receives fewer samples, leading to increased variance in the statistical estimate and a slight decrease in performance. Overall, the method is robust to the number of clusters within a certain range and does not require extremely fine-tuning of parameters.
>
> **Table 2: Ablation of Feature Combinations**
>
> | Feature Set | GSM8K | MATH | AIME24 | MATH500 | **AVG.** |
> | --- | --- | --- | --- | --- | --- |
> | Semantic Only | 82.56% | 16.34% | 20.32% | 42.67% | 40.47% |
> | Difficulty Only | 82.79% | 19.14% | 21.93% | **47.33%** | 42.80% |
> | Lexical + Syntactic | 80.14% | 15.42% | 19.79% | 42.67% | 39.50% |
> | **All Feature** | **83.42%** | **19.36%** | **24.60%** | 46.67% | **43.51%** |
>
> The results show that fusing all features, compared to using a single feature, consistently improves performance, particularly on harder tasks (e.g., AIME24).
>
> Regarding the workload of cluster construction: Feature extraction can be automatically completed using existing tool libraries (NLTK, textstat, BGE-M3) based on the prompt text in the data, without the need for feature selection or assigning different weights. The hyperparameters of clustering have a limited impact on the final performance within a certain range, therefore hyperparameter search is unnecessary. The overall engineering workload is manageable.
>
> To further demonstrate that this framework is not limited to the mathematical domain and does not rely on domain-specific feature engineering, we directly transfer our method to a code generation task (MBPP dataset):
>
> Table 3: Performance on MBPP
>
> | Methods | Ours | AdaRFT | Random | DUMP |
> | --- | --- | --- | --- | --- |
> | Acc. | **60.67%** | 59.26% | 53.33% | 54.27% |
>
> The results show that, without targeted code feature design, our method still outperforms the baseline, demonstrating the framework's generality.

---

> ### Author Response · Authors · 2025-11-24
> **Response to weakness**
>
> ## W1. About Innovation
>
> While Influence Function and MAB are existing tools, the core contribution of this paper lies in **solving the "dynamic distribution" problem unique to RLHF**. Unlike supervised fine-tuning (SFT), the "value" of data in RLHF fluctuates dramatically with policy updates (e.g., the training value of a sample decreases after it changes from "difficult" to "easy").
>
> Existing static sampling methods fail to capture this, while methods based solely on difficulty (such as AdaRFT) easily lead to sample homogenization. We transformed the Influence Function into **Cluster-level Aggregation (reducing variance)** and **Sliding Window (capturing dynamism)**, and combined it with Bandit for exploration-based balancing. This is a complete solution tailored to the pain points of RLHF, rather than a simple stacking of modules.
>
> ## W2. Regarding Computational Overhead
>
> Regarding the runtime issue you mentioned, we conducted detailed time overhead tests on Llama 3.2-3B-Instruct:
>
> Table 4: Comparison of Time Cost and Accuracy
>
> | Method | GSM8K | MATH | AIME24 | MATH500 | AVG. Time | Overhead | AVG. Acc. |
> | --- | --- | --- | --- | --- | --- | --- | --- |
> | Random | 385.51 | 372.25 | 63.00 | 34.25 | 213.75 | 1.00x | 41.95% |
> | AdaRFT | 352.5 | 511.5 | 78.25 | 35.26 | 244.38 | 1.14x | 38.82% |
> | DUMP | 379.14 | 375.73 | 61.26 | 33.37 | 212.37 | 0.99x | 43.31% |
> | Ours | 425.70 | 419.28 | 72.88 | 39.53 | 239.35 | 1.12x | **43.51%** |
>
> The result shows that our method only increases time overhead by approximately **12%** compared to the simplest Random Sampling.
>
> This is mainly due to our use of cluster-level approximation, which eliminates the need to calculate Influence for each sample individually, and the Influence Score is updated periodically rather than step-wise. Considering the performance improvement (Average Accuracy +2.06%), we believe this additional computational cost is acceptable and efficient.
>
> Thank you again for your valuable feedback; these discussions have helped us demonstrate the robustness and practicality of our method more comprehensively. We have added the experimental data and analysis described above to the appendix and relevant sections of the final version.

---

### Official Review · Reviewer_2YAE · 2025-11-01

**Soundness:** 2
**Presentation:** 2
**Contribution:** 2
**Rating:** 4
**Confidence:** 3

**Summary:**

The paper proposes an adaptive curriculum learning framework for RLHF that dynamically identifies and samples the most useful training data as the model evolves. Training samples are grouped into clusters based on semantic and difficulty-related features, each treated as an arm in an MAB setup. A cluster score derived from influence functions computed over a sliding training window, quantifies each cluster's contribution to recent parameter updates. This influence score acts as the reward in an UCB policy, allowing the scheduler to balance exploitation of high-impact clusters with exploration of underrepresented regions. The result is a distribution-level adaptive curriculum that improves sample efficiency and enhances generalisation in RLHF training for large language models.

**Strengths:**

* Tackle an important problem of model-aware adaptive curriculum
* This paper redefines and operationalises influence functions in RLHF, where none of the classical assumptions hold.
* Treating influence as a reward signal to drive adaptive data selection policy.

**Weaknesses:**

* The reward (influence score) is inherently stochastic and non-stationary (since the model keeps changing), which challenges the standard assumptions of UCB-based bandit algorithms.
*  The clustering step which defines the arms of the bandit introduces an arbitrary, heuristic component that can strongly affect outcomes. The adaptive bandit may optimise efficiently within a poorly chosen partition, but not over the actual underlying data utility distribution. But how these clusters are formed i.e., how many there are and how features are weighted is manually chosen, not learned or validated in a principled way.
* The clustering space is built from proxies of difficulty and semantics, not from model-internal or policy-driven features that truly govern learning efficiency. Because clustering is not updated during training, the model’s changing notion of “semantic similarity” or “difficulty” is ignored, reducing long-term adaptivity.
* The asymptotic form hides a large constant factor: conjugate gradient iterations over Hessian-vector products make influence computation several times costlier than standard RLHF updates, challenging scalability to real LLMs
* Expressing influence cost as $\mathcal{O}(|\theta|)$ is mathematically neat but hides architectural dependencies. In practice, transformer-based LLMs exhibit significant per-layer overhead that can make the constant factor large and hardware-unfriendly.
* While computationally light compared to gradient updates, maintaining per-cluster influence windows and UCB scores incurs non-negligible memory and synchronisation cost which is not reflected in the asymptotic expression.
* The trade-off parameter introduces a new hyper-parameter balancing cost and adaptivity, but its sensitivity and effect on convergence are not theoretically analysed or empirically ablated.

**Questions:**

Kindly refer to the weaknesses.

---

> ### Author Response · Authors · 2025-11-24
> **Response to weakness (1/2)**
>
> Thank you very much for your detailed and constructive comments. We respond to each point below, and provide new experimental evidence and theoretical analysis.
>
> ---
>
> ## W1. Non-stationarity of Influence Score Challenges UCB Assumptions
>
> We completely agree with your point that changes in Model Policy during RLHF cause the Reward (Influence Score) to be non-stationary, but this is the core reason why we introduce the **sliding window mechanism** in this paper.
>
> - **The sliding window maintains a locally stationary environment**: Standard UCB does indeed assume a stationary distribution, but Sliding-Window UCB is a classic variant for handling non-stationary Bandit problems [1]. By retaining the Influence estimate of the most recent $k$ steps within the sliding window, it is not only to "forget" the outdated Reward distribution, but also to control the magnitude of reward changes, maintain a locally stationary environment for Bandit, and enable the Cluster Score to adapt to the current learning state of the model.
> - Research on non-stationary betting arms in [2] shows that when the reward distribution changes slowly, the sliding window UCB can reach the $O(\sqrt{T})$ regret bound, which is a reasonable assumption for asymptotic policy updates in RLHF.
> - Experimental Verification: Our training curves (Fig. 3 of the paper) show that the method does not oscillate due to non-stationarity, but converges faster and has smaller variance than the baselines, indicating that this design can successfully adapt to constantly changing model states without catastrophic instability.
>
> [1] Garivier & Moulines. "On Upper-Confidence Bound Policies for Non-Stationary Bandit Problems." ALT 2011.
>
> [2] Besbes et al. "Stochastic Multi-Armed-Bandit Problem with Non-stationary Rewards." NeurIPS 2014.
>
> ## W2. Clustering Introduces Heuristic Components
>
> This is an important concern. We provide the following clarifications and additional evidence:
>
> ### Feature Validity Verification
>
> Our clustering features are selected mainly from three aspects, and are not arbitrary: semantic consistency, difficulty and lexical/syntactic complexity capture.
>
> Meanwhile, these features are not manually weighted, we use z-score normalization (Section 3.2) to treat all dimensions equally before K-means++.
>
> Therefore, we conducted the following ablation experiments on Llama3.2-3B-Instruct to compare the effects of using only a single type of feature with the full set of features:
>
> Table 1: Feature Set Ablation
>
> | Feature Combination | GSM8K | MATH | AIME24 | MATH500 | Average |  |
> | --- | --- | --- | --- | --- | --- | --- |
> | Semantics Only | 82.56 | 16.34 | 20.32 | 42.67 | 40.47 |  |
> | Difficulty Only | 82.79 | 19.14 | 21.93 | **47.33** | 42.80 |  |
> | Lexical + Syntax Only | 80.14 | 15.42 | 19.79 | 42.67 | 39.50 |  |
> | **All Features** | **83.42** | **19.36** | **24.60** | 46.67 | **43.51** |  |
>
> Experiments show that the performance of any single feature type decreases (up to 4.01%), proving the necessity of our multi-dimensional considerations in feature engineering, which can cover both "difficulty" and "diversity".
>
> ### Robustness of Cluster Size
>
> Your concern about cluster granularity is also reasonable. We tested different numbers of clusters $N$:
>
> Table 2: Num of Cluster Analysis
>
> | Number of Clusters N | GSM8K | MATH | AIME24 | MATH500 | Average |
> | --- | --- | --- | --- | --- | --- |
> | 5 | 83.17 | 19.14 | 24.06 | **46.67** | 43.26 |
> | **10** | **83.42** | 19.36 | **24.60** | **46.67** | **43.51** |
> | 15 | 83.24 | **19.38** | 23.53 | 45.33 | 42.87 |
> | 20 | 82.79 | 19.26 | 22.46 | 41.33 | 41.46 |
>
> The results show that performance is relatively stable between $N=5$ and $N=15$, peaking at $N=10$. This indicates that as long as the clustering granularity is appropriate, the algorithm is not extremely sensitive to specific partitions.
>
> ## W3. Fixed Clustering & Features are Not Policy-Driven
>
> The reviewer pointed out that fixed clustering cannot adapt to the evolving similarity concept of the model. However, it is important to note:
>
> - **Influence scoring already captures this evolution**, and the UCB scheduler automatically adjusts cluster weights based on influence.
> - Dynamic re-clustering incurs significant overhead (re-embedding, re-clustering the entire dataset).
> - More importantly, dynamic clustering can cause the betting arm itself to become catastrophically non-stationary, exacerbating the problems in W1. Our static clusters provide a stable objective, while dynamic influence scores capture changes in learning value.
>
> While clustering features are pre-computed, the **influence function itself is policy-driven**—$\mathcal{I}(x,o)$ depends on the policy gradient $\nabla_\theta \ell$, the Hessian matrix $H_\theta$, and the validation gradient. Clustering provides a **stable structural prior**, while influence scores inject **dynamic policy awareness**. This design avoids the huge computational overhead of re-clustering at every step.

---

> ### Author Response · Authors · 2025-11-24
> **Response to weakness (2/2)**
>
> ## W4-W5: Computational Cost Issues
>
> You pointed out that the asymptotic complexity $O(\theta)$ masks the huge constant-term overhead of Hessian-vector products, which is very insightful. To address this concern, we not only measured the wall-clock time but also conducted a sensitivity analysis on the hyperparameter, i.e. update interval $m$.
>
> The results of running Llama 3.2-3B-Instruct on four datasets are as follows:
>
> Table 3: Average Clock-Time Overhead
>
> | Method | Total Time (minutes)* | Relative Cost |
> | --- | --- | --- |
> | Random Sampling | 213.75 | 1.00× |
> | AdaRFT | 244.38 | 1.14× |
> | DUMP | 212.37 | 0.99× |
> | **Ours** | **239.35** | **1.12×** |
>
> \* Averaged on GSM8K, MATH, AIME24, and MATH500
>
> Table 4: Impact of Update Interval $m$
>
> | $m$ Value | Total Time (minutes) | Relative Cost | Average Accuracy |
> | --- | --- | --- | --- |
> | 0 (per step) | 479.19 | 2.24× | 43.26% |
> | 3 | 292.93 | 1.37× | **44.07%** |
> | **5** | 239.35 | 1.12× | 43.51% |
> | 10 | 232.29 | 1.09× | 41.31% |
>
> From the results in the two tables above, we find that:
>
> - If Influence ($m=0$) is calculated at every step, the overhead is indeed as high as **2.24 times**, which, as you would expect, is expensive in practice.
> - However, by adopting the strategy we suggest in the paper (updating interval $m=5$), the overhead is reduced to **1.12 times** (i.e., an additional 12% of time), while the performance (AVG 43.51%) suffers almost no loss.
>
> The transformer bottleneck pointed out by the reviewer does exist. Our mitigation measures include:
>
> - **Batch-level influence**: Calculating $\mathcal{I}_{d_j}$ only for the sampled batch, not the entire dataset.
> - **Selective computation**: Updating at an interval of $m=5$ steps, balancing cost and adaptability.
>
> The experiments above also demonstrate that our measures achieve a good balance between performance and efficiency. Our method does not require more time to reach the target performance while having a higher performance ceiling, which is acceptable in engineering implementation.
>
> ## W6. UCB Scheduler Memory Overhead
>
> Storage increase due to maintaining the UCB scheduler and sliding windows:
>
> - Storage per cluster: $N \times k$ influence scores (e.g., 10 clusters × 100 windows = 1000 floating-point numbers = 4000 bytes)
> - UCB statistics: $N$ counts and means
>
> These are negligible compared to model parameters (e.g., 3B parameters = 12GB).
>
> ---
>
> ## W7. Hyperparameter $m$ not analyzed
>
> We analyzed the impact of hyperparameter $m$ on performance and efficiency using Llama 3.2-3B-Instruct.
>
> Table 5: Performance vs. $m$
>
> | $m$ (interval) | GSM8K | MATH | AIME24 | MATH500 | AVG. |
> | --- | --- | --- | --- | --- | --- |
> | 0 (every step) | 83.32% | **19.38%** | 22.99% | **47.33%** | 43.26% |
> | 3 | **83.55%** | 19.34% | **26.74%** | 46.67% | **44.07%** |
> | **5 (default)** | 83.42% | 19.36% | 24.60% | 46.67% | 43.51% |
> | 10 | 82.71% | 18.74% | 22.46% | 41.33% | 41.31% |
>
> Table 6: Time(Minutes) vs. $m$
>
> | $m$ (interval) | GSM8K | MATH | AIME24 | MATH500 | AVG. | Overhead |
> | --- | --- | --- | --- | --- | --- | --- |
> | 0 | 863.40 | 836.56 | 139.75 | 77.06 | 479.19 | 2.24× |
> | 3 | 509.77 | 495.21 | 87.06 | 47.70 | 292.93 | 1.37× |
> | **5** | 425.70 | 419.28 | 72.88 | 39.53 | 239.35 | 1.12× |
> | 10 | 413.52 | 399.79 | 68.28 | 40.55 | 232.29 | 1.09× |
>
> The strong performance of $m=3$ (especially reaching 26.74% on the difficult AIME24) suggests that for challenging inference tasks, slightly more frequent updates, despite the higher cost, may be more beneficial. However, increasing $m$ can improve computational overhead with a small sacrifice in performance. Therefore, we set $m=5$ by default to balance performance and efficiency.

---

### Author Response · Authors · 2025-11-24
**General response to all reviewers**

We sincerely thank all the reviewers for their professional and constructive comments. These comments greatly helped us improve the experimental depth and theoretical explanations of the paper.

### Summary of Reviewer Comments

All four reviewers acknowledged the paper's exploration of RLHF data scheduling, but also raised the following main concerns:

1. Computational overhead and efficiency (Reviewer 2YAE, A9MV)
2. Experimental scale and generalization (Reviewer q3ME, TEci)
3. Theoretical assumptions and methodological robustness (Reviewer 2YAE, A9MV)

Reviewers q3ME and A9MV also raised detailed questions regarding the construction of the validation set and the sliding window mechanism.

### Summary of Author Responses

We provided detailed responses to the above issues and added supplementary experiments:

1. Regarding model scalability, we added the experiment using Qwen2.5-7B-Instruct. Results show that on larger models, our method still outperforms the strong baselines, demonstrating its scalability.
2. For domain generalization, experiments on the code generation task (MBPP) were added. Results show that without modifying the feature engineering part, the accuracy reaches 60.67% (the best baseline is 53.33%), demonstrating the framework's cross-domain generality.
3. Regarding computational efficiency, detailed wall-clock time comparisons are provided. By setting the update interval $m=5$ and batch approximation, our method only increases the time cost by about 12% compared to Random sampling, while improving performance, demonstrating cost-effectiveness. We further analyzed the impact of different update intervals $m$ on performance and time.
4. In terms of theory and robustness, we verified the necessity of different combinations of clustering features (semantic/difficulty/syntactic) and the robustness of the number of clusters ($N=5 \sim 15$). Meanwhile, the theoretical assumptions explain that the sliding window is designed to solve the non-stationarity problem of RLHF by maintaining "local stationarity".

We have carefully addressed all the reviewers' concerns and clarified misconceptions about our approach and will respond to each comment in detail. We would also greatly appreciate the opportunity to engage further in a constructive discussion.

---

### Meta-Review · Area_Chair_wSgK · 2025-12-27

**Summary:**

This paper proposes an adaptive curriculum/scheduling framework for RLHF that clusters training prompts (semantic + difficulty-related features) and treats each cluster as a bandit arm. A cluster score is updated using sliding-window influence-function estimates, and a UCB-style scheduler balances exploitation of high-impact clusters with exploration to maintain coverage.

 Reviewers agreed the motivation is timely and the system is reasonably designed; the main concerns were (i) computational overhead/scalability of influence estimation, (ii) reliance on heuristic/hand-crafted clustering, and (iii) limited evaluation scope/model sizes and some ambiguities in validation/sliding-window details.

In the rebuttal, the authors provided additional evidence: (a) runtime/wall-clock overhead analyses showing ~12% overhead under a practical update interval, plus sensitivity to the interval hyperparameter,  (b) scaling to a 7B model (Qwen2.5-7B-Instruct) with consistent gains over baselines, and (c) a non-math code generation task (MBPP) showing improved accuracy without domain-specific feature redesign.

Overall, I find the rebuttal addresses some empirical/scalability questions; remaining weaknesses are primarily about broader generality to preference/RM-style RLHF, degree of novelty (composition of known tools), and the design of the clustering mechanism. I lean to reject the paper.

**Reviewer Concerns:**

Concerns substantially addressed by the rebuttal

- Empirical scale / larger model: Added full results on Qwen2.5-7B-Instruct with improvements over Random and strong baselines, including AIME24 gains.

- Beyond math domain: Added MBPP code-generation results showing improved accuracy over baselines.

- Runtime / overhead clarity: Provided wall-clock comparisons and showed overhead can be controlled (e.g., update interval) while retaining accuracy (+12% over Random in their report).

- Validation set construction: Clarified a 20% hold-out built via stratified sampling across clusters to reduce bias.

- Sliding-window ambiguity: Clarified “last k” as a FIFO queue of recent influence scores per cluster; also clarified that the window stores scores rather than serving as a sampling buffer.

- Heuristic clustering sensitivity: Added ablations on feature combinations and number of clusters, showing relative robustness across N=5–15 and benefits from combining features.

Concerns still outstanding

- Generality to “true” preference/RM-based RLHF: Even with MBPP, evaluations remain mostly objective, benchmark-style tasks; it is still unclear how well influence+cluster-bandit behaves under noisy/subjective reward models or preference feedback settings (as one reviewer explicitly requested).

- Novelty level: The core algorithmic pieces (influence estimates, UCB/bandits, clustering) are standard; the contribution is mainly a system integration tailored to RLHF rather than a new algorithmic result.

- Feature engineering dependence: While the authors argue engineering is manageable, the approach still depends on a fixed feature space and static clustering, which may require adaptation in other domains.

**Reviewer Scores:**

I cannot reliably answer this counterfactual question without putting words in reviewers’ mouths. I will not impute score changes beyond what reviewers explicitly stated in the discussion. I instead provide a faithful synthesis of the discussion outcomes and remaining points of disagreement.

---

### Decision · Program_Chairs · 2026-01-26

Reject